# Individual Tree Species Classification Based on Convolutional Neural Networks and Multitemporal High-Resolution Remote Sensing Images

**DOI:** 10.3390/s22093157

**Published:** 2022-04-20

**Authors:** Xianfei Guo, Hui Li, Linhai Jing, Ping Wang

**Affiliations:** 1College of Geodesy and Geomatics, Shandong University of Science and Technology, Qingdao 266590, China; 201883020021@sdust.edu.cn (X.G.); skd990058@sdust.edu.cn (P.W.); 2International Research Center of Big Data for Sustainable Development Goals, Beijing 100094, China; jinglh@aircas.ac.cn; 3Key Laboratory of Digital Earth Science, Aerospace Information Research Institute, Chinese Academy of Sciences, Beijing 100094, China; 4Hainan Key Laboratory of Earth Observation, Sanya 572029, China

**Keywords:** convolutional neural network, multitemporal high-resolution remote sensing imagery, individual tree species classification

## Abstract

The classification of individual tree species (ITS) is beneficial to forest management and protection. Previous studies in ITS classification that are primarily based on airborne LiDAR and aerial photographs have achieved the highest classification accuracies. However, because of the complex and high cost of data acquisition, it is difficult to apply ITS classification in the classification of large-area forests. High-resolution, satellite remote sensing data have abundant sources and significant application potential in ITS classification. Based on Worldview-3 and Google Earth images, convolutional neural network (CNN) models were employed to improve the classification accuracy of ITS by fully utilizing the feature information contained in different seasonal images. Among the three CNN models, DenseNet yielded better performances than ResNet and GoogLeNet. It offered an OA of 75.1% for seven tree species using only the WorldView-3 image and an OA of 78.1% using the combinations of WorldView-3 and autumn Google Earth images. The results indicated that Google Earth images with suitable temporal detail could be employed as auxiliary data to improve the classification accuracy.

## 1. Introduction

Forest resources are an essential resource that has significance for the sustainable development of human society [1,2]. Forests could improve the living environment of human beings and provide habitats for creatures [3,4]. Individual trees are the main body of forests, and the investigation of individual tree species (ITS) is an essential part of forest resource surveys [5,6,7]. The ITS investigations that include the classification and distribution of the individual trees are helpful for the management and protection of forests [8,9,10]. Remote sensing technology has been widely employed in ITS classification because of its macroscopic, dynamic, and abundant acquisitions [11,12,13]. The remote sensing data utilized for ITS classification mainly include airborne hyperspectral images, airborne LiDAR data, and high-resolution aerial photographs [14,15].

Hyperspectral data have many spectral bands, ranging from visible to near-infrared to shortwave infrared (SWIR) [16]. Airborne hyperspectral data have a higher spatial resolution than those from satellites. Therefore, airborne hyperspectral data have become a reliable data source for ITS classification [17,18,19,20,21,22,23,24,25]. Although ITS classification with airborne hyperspectral images could yield high classification accuracy, the high cost of airborne hyperspectral data limits their application in the ITS classification of large-area forests. Airborne light detection and ranging (LiDAR) technology obtains horizontal and vertical structural information of ground objects by actively transmitting lasers to targets and receiving point cloud, echo information data [26,27]. In most studies, LiDAR data were often combined with hyperspectral data to improve the classification accuracy [28,29,30,31]. However, the cost of airborne LiDAR data was high, and the processing of airborne LiDAR data was complex, which was not suitable for large-scale forest classification [32].

High-resolution aerial images have higher spatial resolution and richer texture information than satellite images [33]. Currently, aerial photographs have been utilized in many studies for ITS classification. Kuzmin et al. [34] classified three dominant tree species, and the OA and kappa coefficient were 81.9% and 0.73, respectively. Scholl et al. [35] used airborne photographs to classify four tree species and obtained a maximum classification accuracy of 69% for a RF. With the advantages of flexibility and convenience, unmanned aerial vehicles (UAVs) have become a popular airborne platform for obtaining aerial photographs [36]. Some scholars have utilized UAV data for ITS classification. For example, Franklin et al. [37] utilized UAV data to classify four tree species. The highest OA achieved by the RF method was 78%. Xu et al. [38] utilized the RF method to classify eight tree species. By comparing different feature combinations, the researchers discovered that the combination of spectral and texture features and structural indicators could obtain the highest accuracy. Because of the multiple convolution structure, a CNN could automatically extract features in deep layers from the input images, which yields high classification scores in classification tasks. Some of the studies have attempted to use CNN models for the classification of tree species. For example, Kattenborn et al. [39] successfully applied CNN models in ITS classification, and the best root mean squared error (RMSE) for two tree species reached 0.099 and 0.111. Cao et al. [40] proposed an improved Res-UNet network, which is a combination of U-Net and ResNet. Res-UNet offered an OA of 87% for six tree species, which was significantly higher than those of U-Net and ResNet. However, the acquisition of aerial photographs depends on the availability of various airborne platforms, which are restricted to meteorological and terrain conditions. This limitation also contributes to low stability and the high cost of aerial photographs [41].

With the development of satellite imaging technology, the resolution of satellite imagery has been continuously improved. For example, the panchromatic image resolution of Worldview-3 data could reach 0.31 m [42,43]. The high spatial resolution provides a new approach for ITS classification. Some scholars have conducted relevant studies. For example, Li et al. [44] compared the classification performances of shadow-removed and shadow-recovered Worldview-2 images. The experimental results showed that shadow-removed imagery could offer higher classification accuracy than shadow-recovered imagery, and the highest OA for five urban tree species reached 80.1%. Deur et al. [45] employed Worldview-3 images for ITS classification in a mixed deciduous forest. The RF method provided the highest OA of 85% for three tree species. As the spectral characteristics of some tree species vary with seasonal changes, some scholars have also combined high-resolution satellite images of different seasons to improve the ITS classification accuracy. For example, Fang et al. [46] employed multitemporal WorldView-3 imagery to classify 19 street tree species at different taxonomic levels. The RF method offered an OA of 61.3% at the species level and an OA of 73.7% at the genus level. Ferreira et al. [47] classified eight tree species utilizing the gray level cooccurrence matrix (GLCM). The producer’s accuracies obtained by WorldView-3 images of wet seasons and dry seasons reached 70% and 68.4%, respectively. Although multitemporal, high-resolution satellite images were utilized, these studies did not explore the potential of CNN-based classification methods. Additionally, multitemporal, high-resolution satellite images, such as Worldview images, are not always available. On the other hand, the cost of multitemporal Worldview images will be greatly increased compared to that of a mono-temporal image. Google Earth could offer multitemporal, high-resolution images in different seasons. The highest resolution of Google Earth images could reach 0.11 m. Furthermore, the acquisition of multitemporal Google Earth images is more accessible than that of Worldview images, which provides an available approach for ITS classification, utilizing the spectral characteristics of trees in different seasons.

In this work, the potential of several CNN models (GoogLeNet, ResNet, and DenseNet) for ITS classification was explored using high-resolution satellite imagery covering a natural forest area. The ITS classification accuracies obtained using a single Worldview-3 image (recorded in summer), two Google Earth images (recorded in spring and autumn, respectively), and the combinations were also compared to explore the potential of Google Earth images used as an auxiliary data source to the Worldview-3 imagery for ITS classification. The results of this work were helpful for selecting appreciative CNN models for ITS classification and also provided a valuable reference for using Google Earth images as an auxiliary data source for ITS classification.

## 2. Materials and Methods

### 2.1. Study Area

The study area is located in the western suburb of Beijing, China (39°58′00″–39°59′30″ N, 116°10′30″–116°12′00″ E) (Figure 1), covering an area of approximately 59.7 km^2^. In the continental monsoon climate area of the temperate zone, the study area has an annual average temperature of 16.2 °C and annual average precipitation of 483.9 mm. With a forest coverage rate above 96%, the study area contains both natural forests and artificial forests. Tree species are rich in the study area [48], and conifers and broad-leaved trees are mixed in most of the study area, resulting in a high canopy density. Seven tree species, including cypress (*Cupressus*), pine (*Pinus*), locust (*Sophora japonica*), maple (*Acer* spp.), oak (*Quercus* L.), ginkgo (*Ginkgo biloba* L.), and goldenrain tree (*Koelreuteria paniculate Laxm.*), were considered in this study. Cypress and pine are evergreen conifers, whereas locust, oak, and ginkgo are deciduous broad-leaved trees.

### 2.2. Data

#### 2.2.1. Worldview-3

Launched on 13 August 2014, the WorldView-3 satellite (Digital Globe, Longmont, Co, USA) provides single-band panchromatic images and multispectral images with eight visible and near-infrared bands and eight shortwave infrared bands [49]. The spatial resolutions of the panchromatic band, eight visible and near-infrared bands, and eight shortwave infrared bands were 0.31 m, 1.24 m, and 3.7 m, respectively. The WorldView-3 image was acquired on 23 June 2018, with a panchromatic band and eight multispectral, visible, and near-infrared bands for ITS classification. Table 1 shows the wavelength range and center of the panchromatic band and eight multispectral visible bands. The seven tree species investigated for the study were exuberant in June. Therefore, the WorldView-3 image was suitable for conducting ITS classification.

First, radiometric calibration and RPC orthorectification of the WorldView-3 images were performed in ENVI 5.3. Second, the eight multispectral bands and a panchromatic band were fused by using the reduced misalignment impact (RMI) fusion method [50] to obtain a fused multispectral image (Figure 2) with a spatial resolution of 0.31 m. The fused image was then used for individual tree crown delineation and individual tree species classification. The RMI fusion method could reduce the spectral distortion of fused dark pixels and enhance the boundaries between two different image targets.

#### 2.2.2. Google Earth

Google Earth is a virtual earth software developed by Google that arranges satellite photographs, aerial photographs, and GIS on a three-dimensional earth model [51]. Google Earth satellite images do not comprise a single source of data but an integration of satellite imagery and aerial photographs. Google Earth could provide satellite images with a spatial resolution to 0.11 m and red, green, and blue bands. The Google Earth images employed in this study were acquired in May 2018 and October 2018, respectively, with a spatial resolution of 0.5 m. As shown in Figure 3, the two images were denoted as I-SprGE and I-AutGE.

#### 2.2.3. Field Sampling Points

Due to the topography in West Mountain, the ITS samples were collected where they were accessible. The field survey dataset contains attributes for each tree, such as the stem coordinates, common species name, crown diameter at breast height, and stem diameter at breast height. The GPS receiver acquired the stem coordinates, and the crown and stem diameters at breast height were measured by a meter ruler. The position accuracy of an individual tree fell within the submeter. In total, 695 trees were sampled, including 195 cypresses, 120 pines, 90 locusts, 70 maples, 100 oaks, 60 ginkgoes, and 60 goldenrain trees.

### 2.3. ITS Sample Sets

Figure 4 showed the construction steps of the ITS sample sets. First, based on the preprocessed image (Figure 4a), the segmentation image (Figure 4b) was acquired by the crown slices from the imagery (CSI) tree crown delineation method (Figure 4①). Second, according to the field investigation data (Figure 4c) and manual interpretation, the tree species were labeled (Figure 4②) on the segmentation result. Third, cut and output (Figure 4③) the ITC images (Figure 4e) based on the minimum outer cut rectangle of every labeled tree and grouped all ITC images according to the category. Last, the data augmentation method (Figure 4④) was employed on the ITC images and the ITS sample sets used for experiments were obtained.

#### 2.3.1. Individual Tree Crown Delineation

The production of ITS samples requires ITC images, and obtaining complete and accurate ITC images is essential. Traditional ITC images were mainly obtained by manual sketching or multiscale segmentation, but these methods had a low automation degree, and the segmentation accuracy was not high. Therefore, the crown slices from imagery (CSI) crown delineation method [52] was employed to automatically delineate the ITCs. First, the brightness and color components of the original image (Figure 4a) were generated and then filtered using morphologically open operation at multiple scale levels to determine candidate tree tops. Then, the marker-controlled watershed segmentation method was employed to detect candidate tree crowns of different sizes. Finally, the candidate tree crowns were merged to generate a tree crown map (Figure 4b). The CSI method was employed in this work as it yielded good performances for natural forests [52]. 

According to distribution (Figure 4c) of sampled trees obtained from the fieldwork, seven tree species were labeled. This made samples of seven tree species and obtained the original ITS sample sets. The original ITS sample sets include 195 specimens of cypress, 120 specimens of pine, 90 specimens of locust, 70 specimens of maple, 100 specimens of oak, 60 specimens of ginkgo, and 60 specimens of goldenrain tree. ITC delineation provided an essential basis for the ITS samples. The labeling of tree species was performed on the ITC delineation result, and every labeled tree was output from the image.

#### 2.3.2. Data Augmentation

To expand the scale of the ITS sample set, the data augmentation method was performed on the original sample sets. Data augmentation was a method to expand the dataset in deep learning methods [53]. The training of CNN models requires a large number of samples, which can significantly improve the ability of CNN models. Data augmentation can obtain several samples based on the original sample set through geometric change, color space transformation, confrontation training, random erasure, etc. This method can extract more information from the original dataset to obtain an enhanced dataset that is more comprehensive than the original dataset. Because of the limitation of the available ITS samples, data augmentation was employed to expand the sample set. Five geometric transformation methods were employed, namely, 90° rotation, 180° rotation, 270° rotation, horizontal inversion, and vertical inversion, on the original ITS sample set. The sample set was expanded to six times the original sample set. The sample number of each tree species is shown in Table 2.

#### 2.3.3. Remote Sensing Imagery Sample Set of ITS

To fully utilize the spectral characteristics of trees in different seasons, the Worldview-3 image and Google Earth images were combined to obtain three merged images, which were denoted as WV3SprGE, WV3AutGE, and WV3SprAutGE. Therefore, with three single images, six images were utilized for ITS classification. Based on the three single images (Worldview-3, I-SprGE, I-AutGE) and three merged images (WV3SprGE, WV3AutGE, and WV3SprAutGE), six sample sets were constructed for ITS classification. The six sample sets were denoted as WV3, SprGE AutGE, WV3SprGE, WV3AutGE, and WV3SprAutGE. All six sample sets contained cypress, pine, locust, maple, oak, ginkgo, and goldenrain tree, with a total of 4170 samples. The six sample sets contained the same tree species and the same number of samples for each tree species, and all samples were resized to 32 × 32. The sample sets were randomly divided into a training set, validation set, and testing set. As shown in Table 2, the ratio of the training set, validation set, and testing set was 3:1:1.

### 2.4. Convolutional Neural Networks

The CNN is a kind of deep neural network with convolutional structures. The multilayer, neural network structure could thoroughly learn and extract the features in the image for image classification. Commonly employed CNNs, including GoogLeNet, ResNet, and DenseNet, were chosen for ITS classification.

#### 2.4.1. GoogLeNet

GoogLeNet was proposed by Szegedy et al. in 2014. GoogLeNet won first place in the 2014 ImageNet Large-scale Visual Recognition Challenge (ILSVRC). To maintain the sparsity of the network structure and utilize the high computational performance of the dense matrix, GoogLeNet designed the inception module [54]. To adapt to the samples used in the experiments, the original GoogLeNet, was adjusted mainly using a convolutional layer that does not change the image size to replace two convolutional layers and two maxpooling layers. The reason for this change was that the input image size of the original GoogLeNet was 227 × 227, and the sample size of this study was 32 × 32, thus this part was adjusted. The structure of GoogLeNet is shown in Figure 5. This study utilized the inception module; note that Inception_V2 was used in this study. The specific structure diagram is shown in Figure 6.

#### 2.4.2. ResNet

Designed and proposed by He et al. in 2015, the ResNet model won the title of 2015 ILSVRC classification task champion. Traditional convolutional networks or fully connected networks have problems such as loss and wastage of information. They also cause gradients to disappear or explode, resulting in deep networks being unable to train. The core of ResNet was the design of the residual block. The residual block introduced an identity skip connection, which relieved the problem of vanishing gradients in deep neural networks and allowed the network to learn more features in images [55].

ResNet with a depth of 34 (ResNet_34) was selected for classification experiments. The input of the CNN models are tree crown sample images with a size of 32 × 32, which is relatively small. When the sizes of the output features of a pooling layer were reduced to 1 × 1, no additional features could be learned by the following layers. Thus, ResNet models with a relatively shallow network, such as ResNet models with depths of 18, 34, and 50, were suggested for ITS classification by Li et al. [56]. They also compared the performances of the three ResNet models for ITS classification using WorldView-2 imagery, and ResNet_34 offered the highest classification accuracy. Therefore, ResNet_34 was selected for ITS classification. The structure of ResNet_34 is shown in Figure 7, and the residual module structure is shown in Figure 8.

#### 2.4.3. DenseNet

DenseNet was proposed in 2017 and won the best paper award of CVPR 2017. DenseNet’s designer proposed a dense connection mechanism: all layers were connected [57]. Each layer will accept all the previous layers as its additional input. In DenseNet, each layer was dimensionally connected with all previous layers and applied as the input of the next layer. For an L-layer network, DenseNet contained a total of L(L+1)/2 connections, which was a denser connection method than ResNet.

DenseNet with a depth of 40 (DenseNet_40) was employed for this study. The structure of DenseNet_40 is shown in Figure 9, and the dense block structure is shown in Figure 10.

#### 2.4.4. Model Training

The deep learning environment was constructed based on Tensorflow2.0.0 and Keras2.3.0. All CNN models were trained in the same deep learning environment.

As introduced in Section 2.3.3, each samples’ image was resized as 32 × 32, and the ratio of the training set and validation set was 3:1. In the settings of training parameters, for avoiding the overfitting, the maximum epoch value was set as 500. If the loss did not decrease after 10 epochs, the training was stopped and the weights that provided the best validation accuracy were saved; the initial learning rate was set to 1 × 10^−4^, the learning rate factor was set to 1 × 10^−3^, and the offline setting of the learning rate was 5 × 10^−7^. Three CNN models utilized for classification experiments were trained by adjusting the parameters during the training process, and the CNN models which achieved the highest accuracy were saved, employing the CNN models saved to the test dataset for acquiring the classification accuracies.

### 2.5. Random Forests

For comparing the classification accuracy obtained by using CNN models, we also performed ITS classification experiments with the RF method. Before the CNNs were applied in image classification, the RF method was one of the most commonly used machine learning methods and was widely applied in image classification [58,59]. As with experiments using CNN models, six ITS sample sets were also employed with the RF method for ITS classification. Furthermore, the classification accuracy acquired by RF and CNN models was compared and analyzed.

### 2.6. Accuracy Metrics

Six metrics, including overall accuracy (OA), precision, recall, F1-measure, user accuracy (UA), and producer accuracy (PA), were utilized to evaluate the ITS classification accuracy. Among them, OA, precision, recall, and F1-measure were utilized to evaluate the overall classification results, whereas UA and PA were employed for the comparisons of each tree species.

The OA is calculated through the total number of correctly classified samples divided by the total number of samples [60]. Precision is the proportion, which truly belongs to a specific class divided by all those classified as that specific class. Recall is also the proportion, which is the number of classified as a specific class divided by all the samples that truly belong to that class [61]. The F1-measure (F1) is related to precision and recall, the calculation formula of F1-measure is shown in Equation (1).
(1)$$F1=2*\frac{\mathrm{Precision}*\mathrm{Recall}}{\mathrm{Precision}+\mathrm{Recall}}$$

## 3. Results

### 3.1. Overall Classification Accuracy

The classification metrics including the overall accuracy, precision, recall, and F1- measure of the three CNN models (GoogLeNet, ResNet_34, and DenseNet_40) based on the six sample sets (WV3, SprGE AutGE, WV3SprGE, WV3AutGE, and WV3SprAutGE) are shown in Figure 11 and Table 3.

#### 3.1.1. Comparison of the Classification Accuracies of Worldview-3 and Google Earth Images

Figure 11 and Table 3 show the overall accuracies obtained of the three models using the six sample sets. Among the three sample sets (SprGE, AutGE, and WV3) constructed by a single image, the WV3 sample set offered OA, precision, recall, and F1-measure values that are significantly higher than those of the SprGE and AutGE sample sets. For the WV3 sample set, DenseNet_40 provided the highest OA value, precision value, recall value, and F1-measure value, which were 75.1%, 0.72, 0.73, and 0.72, respectively. In contrast, the highest OA, precision, recall, and F1-measure values obtained by the AutGE sample set were 48.9%, 0.46, 0.43, and 0.43, respectively. In addition, the classification results obtained with the AutGE sample set were slightly better than those obtained with the SprGE sample set, and the highest OA and precision obtained with the AutGE sample set were 48.9% and 0.46, respectively, which were 6.1% and 0.03 higher than those of the SprGE sample set.

Among the three combined sample sets (WV3SprGE, WV3AutGE, and WV3SprAutGE), the WV3AutGE sample set yielded the highest OA, precision, recall, and F1-measure values. By observing the classification results of WV3 and WV3SprGE using the same three models, it was found that the classification accuracy of WV3 was higher than that of WV3SprGE. WV3SprGE compared to WV3, the OAs yielded by GoogLeNet, ResNet, and DenseNet decreased by 1.3%, 3.9%, and 4.6%. The precision values yielded by GoogLeNet, ResNet, and DenseNet decreased by 0.06, 0.04, and 0.03. The recall values obtained by GoogLeNet did not change while those obtained by ResNet and DenseNet decreased. Moreover, the F1-measure values yielded by GoogLeNet, ResNet, and DenseNet also decreased by 0.01, 0.05, and 0.04, respectively. In contrast, the classification accuracies obtained from the WV3AutGE sample set were higher than those obtained from the WV3 sample set. The OAs yielded by GoogLeNet, ResNet, and DenseNet improved by 15.1%, 4.8%, and 3%, respectively. The precision values yielded by GoogLeNet, ResNet, and DenseNet improved by 0.16, 0.11, and 0.08, respectively. The recall values yielded by GoogLeNet, ResNet, and DenseNet increased by 0.19, 0.04, and 0.03, respectively. In addition, the F1-measure values yielded by GoogLeNet, ResNet, and DenseNet also increased by 0.21, 0.06, and 0.06, respectively. The WV3SprAutGE sample set achieved an improvement in OA compared to the WV3 sample set, while compared to the WV3AutGE sample set, only ResNet_34 achieved an improvement, and the other two CNN models obtained a slight decrease in classification accuracies.

#### 3.1.2. Comparison of Different CNN Models

Three CNN models including GoogLeNet, ResNet_34, and DenseNet_40 were employed for ITS classification. Figure 11 and Table 3 show the OA, precision, recall, and F1-measure values obtained by the three models for the WV3, WV3SprGE, WV3AutGE, and WV3SprAutGE sample sets. For the WV3 and WV3SprGE sample sets, DenseNet_40 offered the highest OA, precision, recall, and F1-measure values, followed by ResNet_34 and GoogLeNet. The OA yielded by DenseNet_40 using WV3 and WV3SprGE reached 75.1% and 70.5%, respectively, whereas the precision values were 0.72 and 0.69, respectively. The recall values using DenseNet_40 for WV3 and WV3SprGE reached 0.73 and 0.68, respectively, whereas, the F1-measure values also reached 0.72 and 0.68, respectively. The performance of GoogLeNet was relatively poor among the three CNN models. The OA values acquired by GoogLeNet for WV3 and WV3SprGE were 62.2% and 60.9%, respectively, whereas the precision values offered by GoogLeNet were 0.64 and 0.58, respectively. In addition, the recall and F1-measure values were also inferior to the other two CNN models. 

For the WV3AutGE and WV3SprAutGE sample sets, DenseNet_40 also provided the highest accuracies, followed by GoogLeNet and ResNet_34. For WV3AutGE and WV3SprAutGE, the OAs yielded by DenseNet_40 were 78.1% and 76.5%, respectively, whereas the precision values of DenseNet_40 were 0.8 and 0.78, respectively. The recall values yielded by DenseNet_40 were 0.76 and 0.75, respectively, whereas the F1-measure values yielded by DenseNet_40 were 0.78 and 0.76, respectively. ResNet_34 did not provide high accuracies in experiments utilizing WV3AutGE and WV3SprAutGE, the OAs offered by ResNet_34 were 72.3% and 74.2%, respectively. In the other three accuracy metrics, ResNet_34 also did not provide higher values than DenseNet_40.

By observing the classification results, it was evident that the performances of CNN models were different. Each CNN model had its own unique structure design, which led to the features extracted by model training being various. The various features brought about the difference in classification performance when distinguishing similar tree species or sample sets.

#### 3.1.3. Comparison of the Classification Accuracies of CNN Models and RF

As shown in Figure 11 and Table 3, the highest OA, precision, recall, and F1-measure values obtained by the RF method were 62.2%, 0.61, 0.62, and 0.61, respectively. Among the six sample sets, WV3AutGE yielded the best performance, followed by WV3SprAutGE, WV3, WV3SprGE, AutGE, and SprGE.

By comparing the classification accuracies obtained by the three CNN models and RF, it was evident that the CNN models offered higher classification performance than RF. CNN models could extract features in deep layers from several ITS images; more and deeper features of ITS images made it easier to distinguish different tree species. It can be seen from Figure 11, the classification accuracies of CNN models were higher than the RF method. The highest OA achieved by the CNN models were 78.1%, which is 15.9% higher than that of the RF method. Moreover, the highest recall and F1-measure acquired by CNN models were 0.76 and 0.78, respectively, which are 0.14 and 0.17 higher than the RF method. For each of the six sample sets, the CNN models provided higher accuracy than the RF method.

### 3.2. Classification Accuracy of Tree Species

The UAs and PAs of the seven tree species for the three CNN models (WV3, WV3SprGE, WV3AutGE, and WV3SprAutGE) are shown in Figure 12. Because of the low classification accuracy, the PAs and UAs of each tree species obtained using the sample set SprGE and AutGE were not analyzed.

As shown in Figure 12a–n, the classification accuracies of oak were higher than those of other tree species. The PAs of oak exceeded 90% and could even reach 100%, and the UAs also exceeded 80%. The UAs and PAs of cypress were slightly lower than those of oak. The PAs were approximately 70%, and the UAs exceeded 70%. The classification accuracies of pine were deficient to those of cypress and oak. Among the three CNN models, the experiment using DenseNet_40 acquired higher classification accuracies, and both the PAs and UAs yielded by DenseNet_40 could exceed 70%. The classification accuracies of locust were not as high as cypress and pine. In the experiments using the WV3 sample set, the PAs offered by ResNet and DenseNet exceeded 75%, and UAs offered by ResNet and DenseNet exceeded 60%. The classification accuracies of maple and ginkgo offered by different sample sets differed widely. The UAs and PAs of maple acquired by experiments using WV3AutGE and WV3SprAutGE were much higher than those of WV3 and WV3SprGE. The PAs of ginkgo yielded from experiments using WV3AutGE and WV3SprAutGE were much higher than those of WV3 and WV3SprGE, but the differences in UA were not as large as those in PA. Of the seven trees, the PAs and UAs of goldenrain tree were the lowest, with none of the UAs exceeding 60%; the PA exceeded 70% in only a few experiments.

The differences between the classification accuracies of the seven tree species were also related to the spectral features of the seven tree species. Figure 13 showed the spectral curve of the seven tree species in the 8-band Worldview-3 imagery. The differences between the seven tree species can be observed in the red edge, NIR-1, and NIR-2 bands, especially in the NIR-1 band. It was evident that the spectral curves of the cypress and oak were significantly different from those of other tree species, which contributes to the higher accuracies of the two species than the other species. Meanwhile, the spectral differences between the ginkgo and goldenrain tree were minor, which increased the difficulty of distinguishing them. This contributes to the low classification accuracies of the goldenrain tree.

In addition, by observing the classification results of seven tree species, it was found that the accuracies of maple and gingko yielded by WV3AutGE and WV3SprAutGE were significantly higher than those of WV3 and WV3SprGE. The leaves of maple and ginkgo presented different colors in summer and autumn, respectively; thus, the spectral characteristics presented by images differed. However, a mono-temporal image could not contain the spectral characteristics in two seasons. By combining the images in summer and autumn, the merged image could contain the spectral characteristics of tree species in two seasons, which made it easier to distinguish ginkgo and maple, thus the classification accuracies of maple and gingko could be improved. Compared with the classification accuracies yielded by WV3, the classification accuracies achieved by WV3SprGE did not increase. The results showed that the spectral information contained in the spring Google Earth image was not helpful to improve the classification accuracies.

## 4. Discussion

In the ITS classification experiments, six ITS sample sets constructed using a Worldview-3 image and two Google Earth images, were classified using three CNN models to explore the potential of several typical CNN models and multitemporal, high-resolution, satellite imagery for ITS classification.

In this study, a Worldview-3 image and two Google Earth images were utilized for ITS classification. In the experiments using the SprGE, AutGE, and WV3 sample sets, the WV3 sample set offered higher classification accuracies than the others. This is due to the fact that the Worldview-3 image used in the experiments had higher spatial resolution and more spectral bands than the GE images, which is helpful for distinguishing different tree species. Several previous studies also used Worldview imagery for ITS classification. For example, Deur et al. [45] employed Worldview-3 image to classify three tree species in a mixed deciduous forest, which offered the highest OA of 85%. Compared with Deur’s work, a larger number of tree species was considered in this study. Majid et al. [62] employed a Worldview-3 image to classify seven tree species. Of the seven species, the highest classification accuracy was 71%, the lowest was only 35%, while the OA was relatively low. In their work they used 244 samples, whereas a total number of 695 samples was included in this study. A larger number of tree samples could provide more features of the target tree species, which is helpful for improving the accuracy of ITS classification. Therefore, expanding the samples in the sample set was necessary for improving the classification performances.

The forest coverage rate of the study area is also an essential factor influencing the classification accuracy. The research area of this study was located in a forest area with a forest coverage rate of 96%. The classification accuracies offered in this study were close to some studies conducted in urban regions [63,64]. Liu et al. [63] employed the Worldview image to classify seven tree species in the urban area, and the highest classification accuracy reached 77.5%. Pu and Landry [64] employed multitemporal satellite images to classify seven tree species in the urban area, and the highest classification accuracy reached 75%. In the urban area, trees were planted artificially and distributed regularly, which was beneficial for classification. The crowns of trees in urban or low coverage areas were also clearer than those in forest areas with high forest coverage, making it easier to locate and delineate the crowns. In contrast, it was difficult to delineate the crowns in high forest coverage areas. The result of delineation affected the quality of samples, which determined the classification accuracy.

Multitemporal Google Earth images were combined with the Worldview-3 imagery for ITS classification. The results showed that the inclusion of the autumn Google Earth image could improve the classification accuracy. Madonsela et al. [65] utilized multitemporal Worldview-2 images to classify four deciduous tree species and yielded the highest OA of 76.4%. Compared with Madonsela’s study, more tree species were classified, and higher classification accuracies were offered in this study. In Madonsela’s study, the images were acquired in March and April, which belong to spring. The images utilized in this study were acquired in May, June, and October. In autumn, the leaves’ color of some tree species, such as maple and ginkgo, changed regularly, resulting in the the change in spectral information. Therefore, the combination of images in different seasons, especially in summer and autumn, could achieve higher accuracy than the combination of images in one season. In spring, cypress and pine were in the period of florescence, in which the spectral characteristics differed from other seasons. However, the spring Google Earth image and the Worldview-3 image did not acquire a better classification performance. The reason may be that the spectral characteristics of cypress and pine did not change significantly, as in summer and autumn, thus the classification accuracy was not improved. Compared to the images of spring and autumn, the images in autumn were more suitable for application in classification to improve accuracy. 

In addition, DenseNet_40 and ResNet_34 yielded better performances than GoogLeNet, due to the excellent structural design of DenseNet and ResNet_34. All layers of DenseNet were connected, which can enhance the transfer of features and allow the extracted features to be used more effectively. Hartling et al. [66] also utilized DenseNet to classify eight tree species and achieved the highest OA of 82.6%. The forest coverage rate of their study area was 18.2%, which is significantly lower than that of this study. The residual module used by ResNet could reduce the waste and loss of features during the transfer processes, which explains why ResNet could offer high classification accuracy. Cao et al. [40] classified six tree species in a mountain area, and the highest OA offered by ResNet reached 68.3%. Sun et al. [32] also utilized ResNet to classify six tree species in a wetland park, and the highest OA reached 89.78%. Compared to Sun’s study, more tree species were classified in this study. Meanwhile, a larger number of samples was used in Sun’s study, which was another reason for the higher accuracy offered in Sun’s study. Different tree species were considered in different studies, and the spectral differences between the tree species were also one of the factors influencing the classification accuracy. Rezaee et al. [67] utilized VGG-16 to classify four tree species, including red maple, birch, spruce, and pine. The two deciduous broad-leaved species, red maple and birch, yielded higher accuracies than the two evergreen conifers species, spruce and pine. Different CNN models have different structure designs and depths, contributing to the differences in the accuracies obtained using the same sample set. Ferreira et al. [68] compared the performances of three CNN models (ResNet_18, ResNet_50, MobileNetV2), and the results showed that ResNet_18 acquired better performance than ResNet_50 and MobileNetV2. 

## 5. Conclusions

The classification of ITS is significant to the management and protection of forest resources. A detailed and accurate survey of tree species distribution will help forest resource managers manage and utilize forest resources. By employing CNN models for classification in six sample sets, the classification performances of several CNN models (GoogLeNet, ResNet_34, and DenseNet_40) were compared, and the potential of multitemporal high-resolution satellite images in different seasons for ITS classification was also explored. 

The three CNN models in this study offered high classification accuracies, which were higher than those of RF. Among the three CNN models, DenesNet_40 provided the highest classification accuracy, followed by ResNet and GoogLeNet. Therefore, the three CNN models have a great application potential in ITS classification. The utilization of mono-temporal Worldview-3 image and multitemporal Google Earth images offered higher classification accuracy compared to employing a single data source. The classification results demonstrated that the utilization of Google Earth and Worldview-3 images could improve the classification accuracies of some tree species, such as ginkgo and maple. Thus, the overall classification accuracy was improved. In the experiment using Worldview-3 and autumn Google Earth images, the OA reached 78.1%, which was 3% higher than that using Worldview-3 only. Compared with multitemporal, high-resolution, satellite images, the combination of Worldview-3 and Google Earth images improved the classification accuracy and reduced the cost. The employment of Google Earth images could offer an effective way of using features contained in multiple seasons for ITS classification. 

In future ITS classification research, the classification accuracy could be improved in the following aspects: (1) A larger number of samples could improve the ITS classification accuracy. When CNN models are employed for ITS classification, more samples will help the CNN models extract more features of trees for ITS classification, improving classification accuracy. (2) The potential of other low-cost high-resolution satellite imagery used as auxiliary data can also be explored to improve the classification accuracy. For example, time sequence GaoFen-2 (GF-2) and GaoFen-7 (GF-7) imagery can be combined with WorldView-3 imagery for ITS classification in further work. 

## Figures and Tables

**Figure 1 sensors-22-03157-f001:**
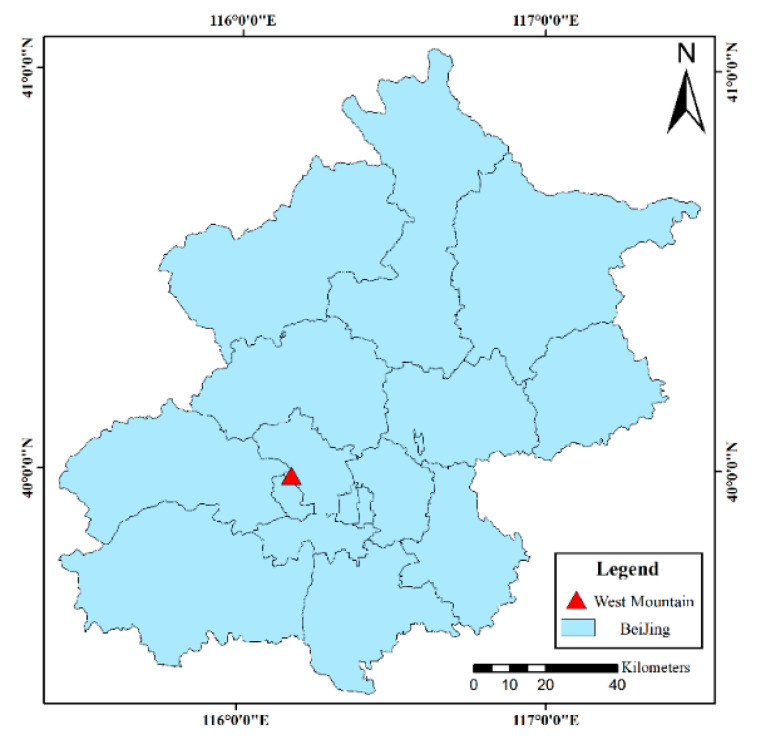
Location of the study area.

**Figure 2 sensors-22-03157-f002:**
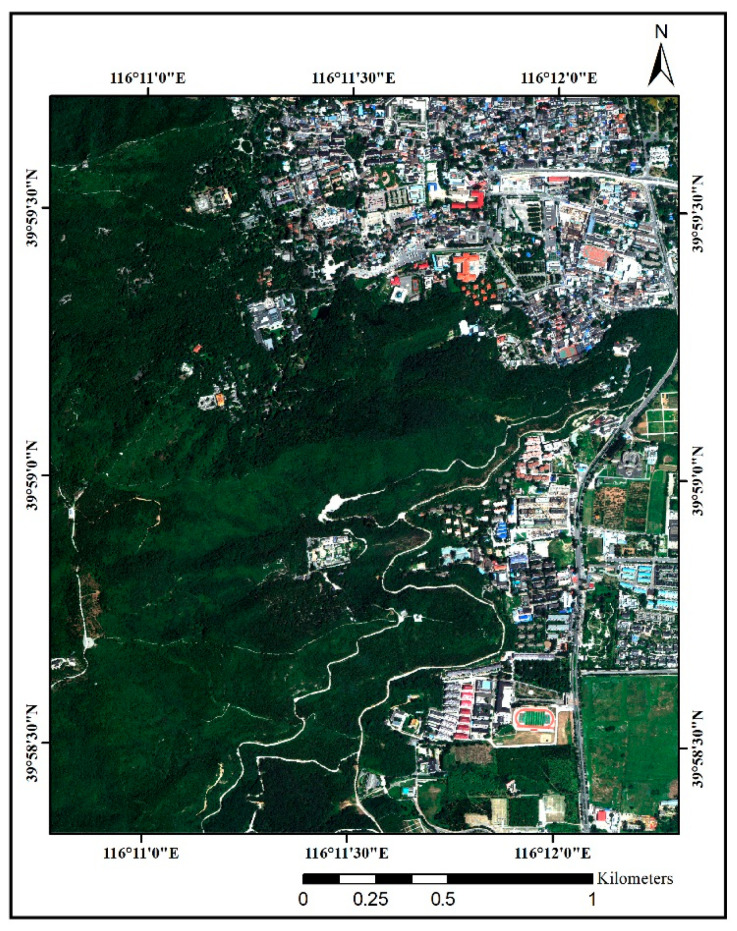
Worldview-3 imagery of West Mountain (The imagery was composed of the red band, green band, and blue band).

**Figure 3 sensors-22-03157-f003:**
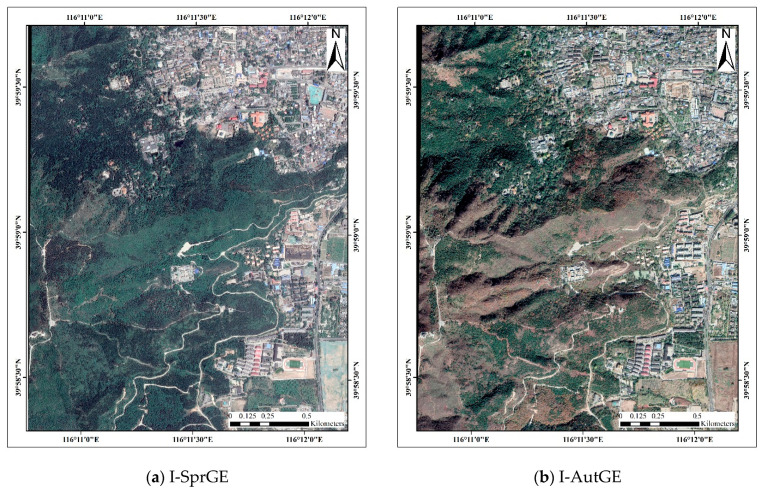
Google Earth images.

**Figure 4 sensors-22-03157-f004:**
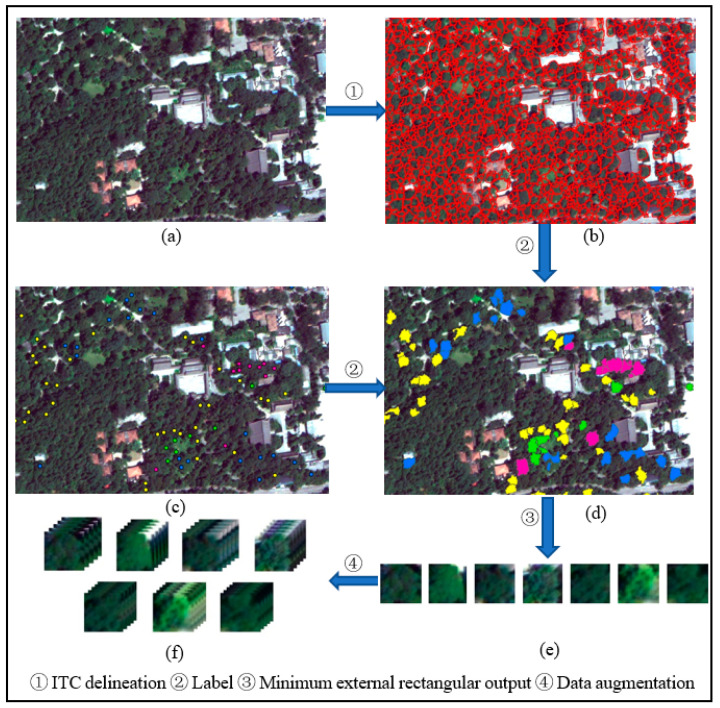
Construction of the ITS sample set (taking Worldview-3 images as an example). (**a**) Original image; (**b**) segmentation image (individual tree crown, ITC) delineation result; (**c**) location of field investigation samples; (**d**) tree species labeling; (**e**) ITS samples; (**f**) ITS samples after data augmentation.

**Figure 5 sensors-22-03157-f005:**
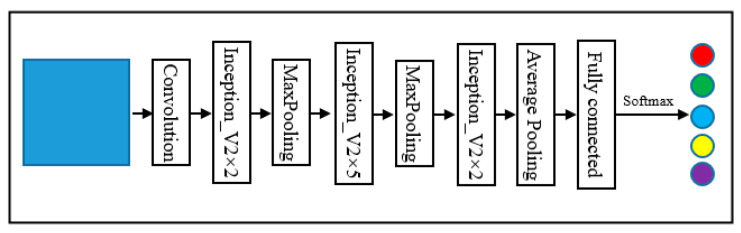
Structure of GoogLeNet.

**Figure 6 sensors-22-03157-f006:**
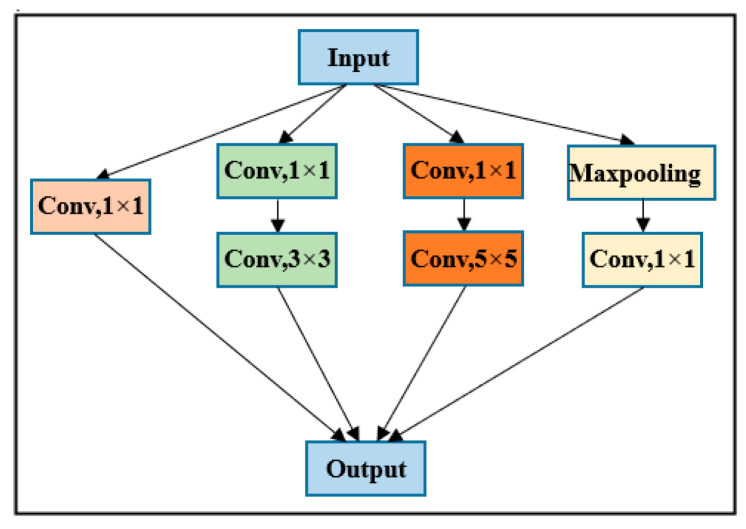
Structure of Inception_V2.

**Figure 7 sensors-22-03157-f007:**
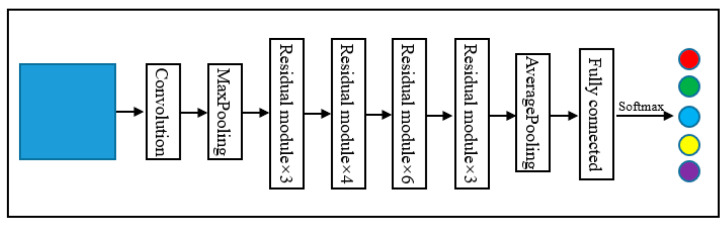
Structure of ResNet_34.

**Figure 8 sensors-22-03157-f008:**
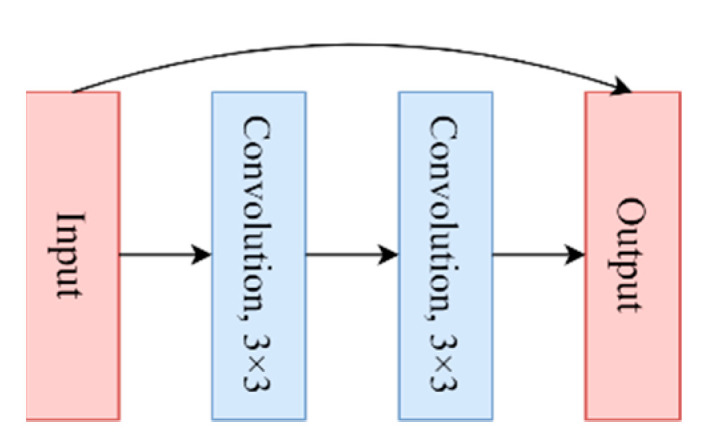
Structure of the residual module.

**Figure 9 sensors-22-03157-f009:**
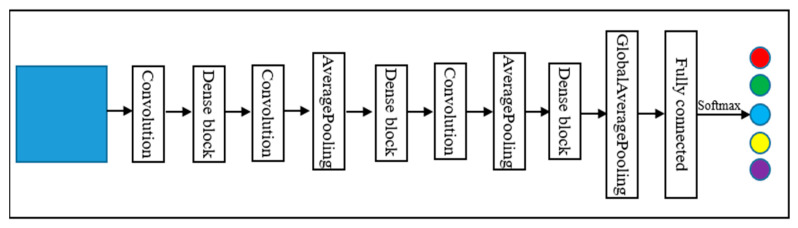
Structure of DenseNet_40.

**Figure 10 sensors-22-03157-f010:**
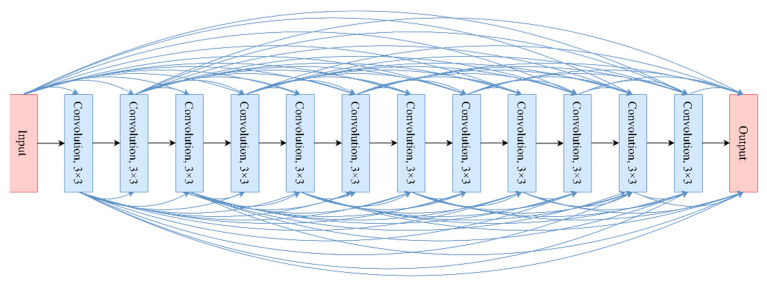
Structure of the dense block.

**Figure 11 sensors-22-03157-f011:**
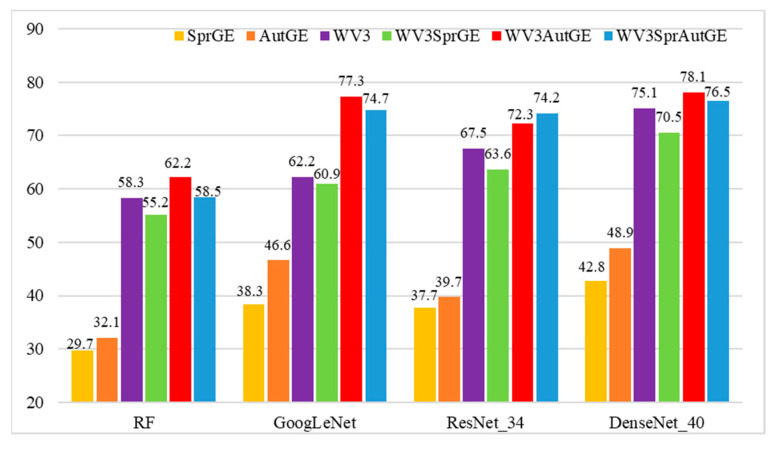
Overall accuracy.

**Figure 12 sensors-22-03157-f012:**
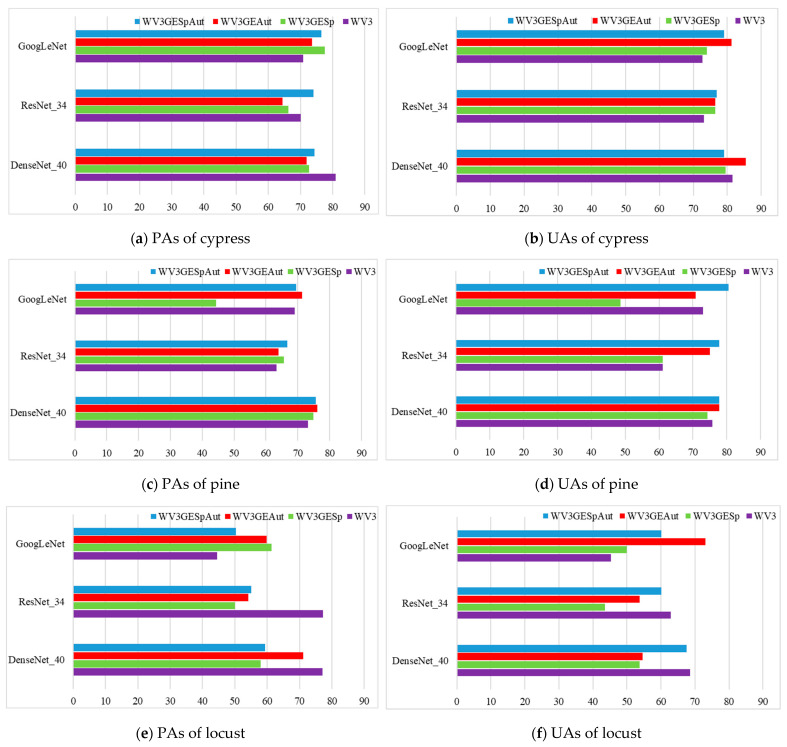
Classification accuracy of seven tree species.

**Figure 13 sensors-22-03157-f013:**
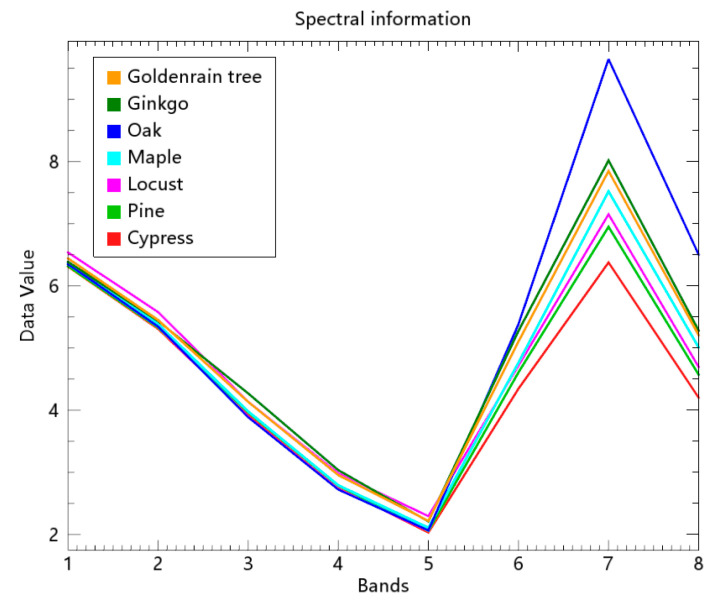
The spectral curve of the seven tree species considered in this work.

**Table 1 sensors-22-03157-t001:** Worldview-3 bands.

Band	Wavelength Range (nm)	Wavelength Center (nm)
Panchromatic	450–800	625
Coastal	400–450	425
Blue	450–510	480
Green	510–580	545
Yellow	585–625	605
Red	630–690	660
Red Edge	705–745	725
NIR-1	770–895	832.5
NIR-2	860–1040	950

**Table 2 sensors-22-03157-t002:** Information of the final ITS sample set.

Name	Number of Samples			Total
	Train	Validation	Test	
Cypress	702	234	234	1170
Pine	432	144	144	720
Locust	324	108	108	540
Maple	252	84	84	420
Oak	360	120	120	600
Ginkgo	216	72	72	360
Goldenrain tree	216	72	72	360
-	2502	834	834	4170

**Table 3 sensors-22-03157-t003:** Classification results (precision, recall, and F1-measure).

Sample Set		SprGE	AutGE	WV3	WV3SprGE	WV3AutGE	WV3SprAutGE
Method	Metrics						
RF	Precision	0.27	0.31	0.57	0.54	0.61	0.57
	Recall	0.28	0.31	0.58	0.54	0.62	0.57
	F1	0.27	0.31	0.57	0.55	0.61	0.57
GoogLeNet	Precision	0.29	0.43	0.64	0.58	0.80	0.75
	Recall	0.33	0.39	0.57	0.57	0.76	0.72
	F1	0.30	0.39	0.57	0.56	0.78	0.73
ResNet_34	Precision	0.34	0.36	0.66	0.62	0.77	0.75
	Recall	0.34	0.34	0.66	0.60	0.70	0.73
	F1	0.34	0.34	0.66	0.61	0.72	0.74
DenseNet_40	Precision	0.43	0.46	0.72	0.69	0.80	0.78
	Recall	0.39	0.43	0.73	0.68	0.76	0.75
	F1	0.40	0.43	0.72	0.68	0.78	0.76

## Data Availability

Not applicable.

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
