# Peer review of "Individual Tree Species Classification Based on Convolutional Neural Networks and Multitemporal High-Resolution Remote Sensing Images"

_sensors, 2022, doi:10.3390/s22093157_

Round 1

Reviewer 1 Report

Individual tree species classification based on convolutional neural networks and multitemporal high-resolution remote sensing images

This manuscript uses high-resolution multitemporal satellite imagery (i.e., WorldView-3 and Google Earth imagery) for individual tree species (ITS) classification. The Authors employed convolutional neural network (CNN) models on WorldView-3 and Google Earth imagery in different seasons for seven tree species. The research proved CNNs as a great application potential in individual tree species (ITS) classification.

Review summary

The topic of this manuscript is very actual and interesting for the readers, but it does not provide significant scientific contributions in order to be published in this Journal. My main concerns are, as follows:

  • Abstract: this Section looks like it has not been finished
  • Introduction: the Authors have explained in detail, but unnecessary, remote sensing data such as hyperspectral data (LN 35 – 55), LiDAR data (LN 56 – 71), and high-resolution aerial imagery (LN 72 – 94) with the results of the similar research. However, these results/research are not important for this research since this research only used high resolution satellite imagery
  • LN 160: can the Authors explain the purpose of using the panchromatic band?
  • Section 2.3.1. is currently written as a tutorial, not as a scientific paper
  • Section 2.4. is written in detail about established Convolutional neural networks (CNNs), whereas Section 2.5 (the most important part about evaluating/comparing the research) is only enumerated
  • Section Results starts with four Figures and then starts the description of the results, but more important are the metrics Recall and Precision, usually they are reported for land-cover classes individually, and in Fig.11 (b) and (c) they are single measure, but what is the meaning of it? Can the Authors explain?
  • LN 414: Figure??
  • Figure 12 is very hard to follow
  • Again, one of the scientific contributions from Discussion is (LN 479 – 481): „The Worldview-3 image had more bands than the Google Earth image (..)“ By merging multitemporal images in different seasons covering a forest area, our study classified more tree species (seven dominant tree species) in the forest areas with a high tree canopy cover rate.“ This is not enough
  • End of Conclusion: „Higher-resolution remote sensing images will improve the accuracy of individual tree crown delineation, thus improving ITS classification accuracy“. In the research commercial satellite imagery was used with spatial resoultion of 0.31 m (RMI fusion) and Google Earth was 0.11. And at the end of the research, the Authors want in the future use imagery with even higher spatial resolution??

Overall, I think that the research design of this manuscript is very good and actual, but still, the manuscript is not suitable for publishing in this Journal.

Reviewer 2 Report

The manuscript presents an interesting study related to the use of multitemporal high-resolution remote sensing images for Individual tree species classification. The manuscript needs to be improved, in particular with regard to a better explanation of the obtained results.

Note to the manuscript:

  • lines 338-342 - It is good to give the formulas for determining the parameters used to assess the accuracy of the classification and a brief description of what each of them shows.
  • Figure 11 - The results presented in Figure 11 show the same trend for each of the parameters. They can be presented in a table and illustrated with a single graph.
  • Most of the text in the results section directly retells the data in Figure 11, without explaining the reason for the difference in the accuracy of determining the individual tree species with different models for the same dataset.
  • There is an insufficient explanation for the difference in the accuracy of determining the individual species with the same method and dataset. It would be interesting to provide information on incorrectly classified samples - are they from a particular region or are they evenly distributed, etc.? In addition, providing information on the spectral data for each tree species studied (red, green, and blue coordinates from Google Earth images, Worldview-3 bars) can help explain the results.

Reviewer 3 Report

I have carefully read MS titled,  "Individual tree species classification based on convolutional neural networks and multitemporal high-resolution remote sensing images", which was submitted for consideration in the Sensors (MDPI).

In this paper, the Authors constructed several ITS sample sets based on Worldview-3 and Google Earth images for ITS classification research. Additionally, they conducted the experiments using CNN models and sample sets. The authors explored the combination of a Worldview-3 image and two Google Earth images by comparing the classification accuracies achieved by single images and merged images. They also explored the potential of several typical CNN models (GoogLeNet, ResNet, and DenseNet) for ITS classification using a single Worldview-3 image and combined images.

The paper is in general well written and well-illustrated, and logically structured, while at the same time adequately concise. The title clearly describes the contents of the paper. The abstract looks good and contains all the important information, it encapsulates the entire study (a bit of introduction, aim, result and outcome). However, the introduction should be changed, completed, and stylistically corrected. I believe that the Materials and Methods section is well structured and scientifically sound. Literature reviews in the discussion section of the manuscript are good. The manuscript is well concluded as the main outcomes are well captured with some recommendations. The figures and tables are correct. Although the language is not bad, I feel that editing the text by a native English speaker would help and improve the readability of the paper.

My comments mostly relate to relatively minor issues of interpretation and writing. These comments do not influence a positive impression from the article.

Specific comments:

Lines 26-34: The first paragraph of this article requires rephrasing. Please add additional citations.

Line 38: Dian et al. - please add the number [x] in the list of references

Line 40: Dalponte et al. - as above, please add the number [x] in the list of references

Line 43: Maschler et al. – as above, please complete the reference number

Line 44: - as above - Raczko et al.

Line 46: The results or Raczko et al.? - please specify

Line 47: correct: Nezami et al. [19]…

Line 49: correct: Zhang et al. [20] …

Line 50: correct: Fassnacht et al. [21] …

Line 52: correct: Harrison et al. [22]

Line 56: Please explain LiDAR abbreviation, i.e. Light Detection and Ranging

Line 59: Zhao et al. – number of references needed

Line 62: Marrs et al. – number of references needed

Line 66: Ballanti et al. - number of references needed

Line 74: corect Kuzmin et al. [30]

Line 75: Scholl et al. [31]

Line 80: Franklin et al. - number of references needed

Line 81: similar problem, Xu et al. - number of references needed

Line 87: Kattenborn et al. [35]

Line 89: Cao et al. [36]

Lines 98-100: Sentences require rephrasing

Line 99: Lie et al. [?]

Line 103: Deur et al. [?]

General: please check citations throughout the paper

Line 139: Please complete the Latin names of the taxa, e.g. Cypress (Cupressus), Koelreuteria paniculate - the Latin name of this species should be written in italics

Line 141: Robur? - Did the authors mean Quercus?  This genus is classified to the family Fagaceae! Deciduous conifers occur in five genera (Larix, Pseudolarix, Glyptostrobus, Metasequoia and Taxodium)

Line 170: Google Earth is a geobrowser …

Line 188: … and 60 specimens of Koelreuteria paniculate?

Line 225: Koelreuteria paniculate - the Latin name of this species is written with a capital letter and italics

In general: The headings of the subsections are written in italics

line 266-267: Sentence requires rephrasing

Line 272: please explain what is a “STEM” module

Line 344: Results: At the very beginning of the paragraph, please add a sentence to explain the charts.

Reviewer 4 Report

Proper management of forest resources requires making many decisions from the people managing them. Recently, due to the intensive technological development, data provided by various types of satellites have been very helpful. A lot of information about single tree segmentation or the species structure of stands, using the analysis of individual tree species (ITS) can be used. With regard to the description of the first element, the publication "Use of LIDAR-based digital terrain model and single tree segmentation data for optimal forest skid trail network" (doi.org/10.3832/ifor1355-007) could be included in this paper.

The article refers to  the use of Worldview-3 and Google Earth images and the use of convolutional neural network (CNN) models to improve the accuracy classification of ITS.

It was written correctly meeting the criteria of a research paper. In the final part of the Introduction I propose to remove the text describing the next stages of the work (lines 123-129). This is a place where the purpose of the research should be clearly defined.

In the methodological part, especially with the reference to Figure 4, there is no need to repeat the information contained in the chart and in the text. This should be written more generally.

The results were presented correctly.  I suggest that Discussion should be written without any subsections, keeping the very general character of the considerations.

Please avoid the statements "in this paper" and "in this work".

The article should be written impersonally, avoiding the words "we", "our", etc.

Round 2

Reviewer 1 Report

Individual tree species classification based on convolutional neural networks and multitemporal high-resolution remote sensing images

The aforementioned manuscript has improved from version_01 to version_02. However, I still have some minor concern, and it is listed, as follow:

  • I think that the Author's did not understand my last comment. 'And at the end of the research, the Authors want in the future use imagery with even higher spatial resolution?' I know that we can get already imagery with even higher resolution than WV3 or GE, but at what cost? Is it really necessary to use even higher resolutions for tree species classification. Trade-off between computational cost and spatial resolution needs to be taken into account

The manuscript has improved between the two versions, and it can be published after this minor corrections.

Reviewer 2 Report

The manuscript has been significantly improved.

However, there is still no explanation for the differences in the accuracy of the classification of individual species using the same database and method. If the information on color coordinates and spectral characteristics for different species is provided (range, mean values), this would explain some of the results. For example, are there species with the same range of values, etc?
